# Acute Myocardial Infarction Hospitalizations between Cold and Hot Seasons in an Island across Tropical and Subtropical Climate Zones—A Population-Based Study

**DOI:** 10.3390/ijerph16152769

**Published:** 2019-08-02

**Authors:** Min-Liang Chu, Chiao-Yu Shih, Tsung-Cheng Hsieh, Han-Lin Chen, Chih-Wei Lee, Jen-Che Hsieh

**Affiliations:** 1Institute of Medical Sciences, Tzu Chi University, Hualien 97004, Taiwan; 2Department of Physical Therapy, Tzu Chi University, Hualien 97004, Taiwan; 3General Education Center, Tzu Chi University of Science and Technology, Hualien 97004, Taiwan; 4Division of Cardiology, Tzu Chi Medical Center, Hualien Tzu Chi Medical Center, Hualien 97004, Taiwan

**Keywords:** acute myocardial infarction, risk triggers, seasonal temperatures, climate zones, individual variables

## Abstract

We investigated the effects of cold and hot seasons on hospital admissions for acute myocardial infarction (AMI) at the junction of tropical and subtropical climate zones. The hospitalization data of 6897 AMI patients from January 1997 to December 2011 were obtained from the database of the National Health Insurance, including date of admission, gender, age, and comorbidities of hypertension, diabetes mellitus (DM), and dyslipidemia. A comparison of AMI prevalence between seasons and the association of season-related AMI occurrences with individual variables were assessed. AMI hospitalizations in the cold season (cold-season-AMIs) were significantly greater than those in the hot season (OR 1.15; 95% CI 1.10–1.21). In the subtropical region, cold-season-AMIs were strongly and significantly associated with the ≥65 years group (OR1.28; 95% CI 1.11 to 1.48). In the tropical region, cold-season-AMIs, in association with dyslipidemia relative to non-dyslipidemia, were significantly strong in the non-DM group (OR 1.45; 95% CI 1.01 to 2.09) but weak in the DM group (OR 0.74; 95% CI 0.55 to 0.99). The cold season shows increased risks for AMI, markedly among the ≥65 years cohort in the subtropical region, and among the patients diagnosed with either DM or dyslipidemia but not both in the tropical region. Age and comorbidity of metabolic dysfunction influence the season-related incidences of AMI in different climatic regions.

## 1. Introduction

Acute myocardial infarctions (AMI) are severe cardiovascular events that are caused by a sudden and critical reduction in blood flow to the coronary arteries. Evidence has indicated that coronary artery occlusion may be influenced not only by long-term chronic physiological risk factors but also by short-term exposure to physical, psychosocial, and environmental temperature triggers [1].

Mortality risks have been reported to be attributable to non-optimum cold and hot temperatures, mostly the contribution of cold, from several cities across multiple climate zones [2]. The health consequences of adverse hot temperatures will become more serious after the impacts of climate change, especially for vulnerable populations [3]. The frequency of AMI occurrences is highly related to seasonal changes in temperature [4]. In the studies from cold and temperate climate zones worldwide, hospital admissions for AMI during the cold seasons were higher compared to the hot seasons [5,6,7,8]. However, the impact of seasonality on hospital admissions for AMI was inconsistent in the studies from different geographic locations within the warm climate zone. An increase in the occurrences of AMI during summer was noted in some warm-climate regions of America [9]. In warm-climate regions of Asia, an absence of seasonality in the occurrences of AMI was claimed in a Taiwanese study [10]. By contrast, a study conducted by Goggins et al. from three warm climate cities in Taiwan and Hong Kong revealed that ambient temperature drops below 24 °C substantially increased the occurrences of AMI [11].

The association between seasonal temperatures and the occurrences of AMI appears to be influenced by various individual variables. Several studies showed that the associations between seasons and the occurrences of AMI were influenced by age and gender. However, the effect of age and gender in association with season on AMI were inconsistent in their findings [4,5,6,10]. A recent study from Hong Kong revealed that diabetes mellitus (DM) patients showed a greater frequency of admission for AMI compared to non-DM patients, markedly in the extreme temperature spectrum of both cold and hot seasons [12]. The inconsistency with regard to age and gender within previous research across multiple climate ranges and the effect on the rates of AMI in association with DM and other comorbidities require elucidation.

The Tropic of Cancer (23°26′12.6″ north of the Equator) separates the island of Taiwan into two climate regions: (1) tropical in the south and (2) subtropical in the north. The purpose of this study was to investigate the influence of cold and hot seasons on the rate of AMI patients admitted to hospital in different climate regions of Taiwan. The AMI hospitalization rate was assessed between cold and hot seasons in subtropical and tropical regions. Individual variables of gender, age, and other comorbidities were also examined for their influence on the season-related incidences of AMI in subtropical and tropical regions.

## 2. Materials and Methods

### 2.1. Data Sources

Since 1995, 99% of residents in Taiwan have been covered by the compulsory National Health Insurance (NHI) system [13]. The National Health Insurance Research Database (NHIRD) contains all registration files and details regarding original claims relating to 1 million beneficiaries from the National Health Insurance (NHI) database for the purpose of empirical studies. Patients’ identities in the database are kept anonymous. Data for this study were obtained from the dataset included in the population-based NHIRD.

### 2.2. Study Cases

The data from 7124 cases with hospitalization records (Hosp. ID) for the first occurrence of AMI (defined by the International Classification of Diseases, Ninth Revision, Clinical Modification (ICD-9-CM) diagnosis codes of 140.xx) from January 1997 to December 2011 were obtained from NHIRD. From this dataset of *n* = 7124, further data were retrieved regarding the date of admission, gender, age, DM (250.x), dyslipidemia (272.x), and hypertension (401.x–405.x).

The island of Taiwan is located in East Asia, roughly 150 km off the southeastern coast of mainland China. Taiwan consists of a main island and several outlying islands. The main island of Taiwan is divided geographically into western and eastern areas by the central mountain range, and climatically into tropical and subtropical regions by the Tropic of Cancer (Figure 1). More than 97% of the population inhabits western Taiwan, with less than 3% in eastern Taiwan and the outlying islands. In order to reduce geographical bias, 209 cases from eastern Taiwan and 18 cases from the outlying islands were excluded. A total of 6897 cases from western Taiwan were included in the analysis. For the purpose of this study, the 6897 cases were categorized into two groups: (1) the subtropical group (*n* = 4604), including the cases in the western counties above the Tropic of Cancer, and (2) the tropical group (*n* = 2293), including the cases in the western counties at and below the Tropic of Cancer.

This study was approved by the Research Ethics Committee at the Buddhist Tzu Chi General Hospital (IRB101-98).

### 2.3. Weather Data and Season

The mean daily temperature from 1981 to 2010 was obtained from the Taiwan Central Weather Bureau. The traditional four seasons in the Northern Hemisphere are defined as follows: spring = 1 March to 31 May, summer = 1 June to 31 August, autumn = 1 September to 30 November, and winter = 1 December to 28/29 February. For the purpose of this study, cold and hot seasons were defined as follows: cold season = 1 November to 30 April, hot season = 1 May to 31 October.

### 2.4. Statistical Analysis

The statistical software package Statistical Product and Service Solutions (SPSS) Version 21.0, was used to analyze demographic data. The Test of Goodness-of-Fit was used to perform a univariate analysis and to examine the proportion of cold season AMI cases to hot season AMI cases in various regions. The student’s *t* test and one-way ANOVA evaluated the differences between cold and hot seasons as well as among the four seasons. The backward logistic regression (LR) method (entry criteria: 0.1; removal criteria: 0.05) was performed to assess the association of AMI prevalence in cold season with: (1) age, (2) gender, and (3) co-morbidities. The estimation of the odds ratios (ORs) of various variables and their 95% confidence intervals (CIs) with the significance level set at 0.05 were calculated.

## 3. Results

### 3.1. Descriptive Statistics

From January 1997 to December 2011, the mean daily temperature was 19.2 ± 2.7 °C in the cold season and 27.3 ± 1.8 °C in the hot season in the subtropical region of western Taiwan, and 21.5 ± 2.7 °C in the cold season and 28.0 ± 1.1 °C in the hot season in the tropical region (Table 1). The mean daily temperatures descended down below 24 °C in November and ascended up to and over 24 °C by the end of April.

There were 6897 cases of out of the 7124 AMI hospitalized patients for the period of January 1997 to December 2011, of which 4604 were within the subtropical region and 2293 in the tropical region. The ratio of men to women was 7:3 (4741/2156) in the total population, 7:3 (3152/1452) in the subtropical group, and 7:3 (1589/704) in the tropical group. The average age range for the total population was 66.3 ± 14.0 years (men = 63.9 ± 13.9 and women = 71.3 ± 12.8). The average age range for the subtropical group was 66.0 ± 14.1 years (men = 63.6 ± 13.9 and women = 71.3 ± 13.0). The average age in the tropical group was 66.5 ± 13.7 years (men = 64.4 ± 13.8 and women = 71.3 ± 12.2). The ratio of age <65 years to age ≥65 years was 4:6 (3047/3850) in the total population, 4:6 (2065/2539) in the subtropical group, and 4:6 (982/1311) in the tropical group. Regarding comorbidity, the percentage of AMI cases with hypertension, diabetes, or dyslipidemia was 62.1%, 35.3%, and 18.2% in the total population, respectively. The subtropical group percentages were 61.8%, 34.3%, and 18.3% and the tropical group percentages were 62.8%, 37.3%, and 17.9%, respectively (Table 2).

### 3.2. Comparison of AMI Hospitalizations between Cold and Hot Seasons

The incidence of AMI, defined as the number of AMI hospitalizations divided by the number of overall hospitalizations, was significantly greater during the cold season as opposed to the hot season for the total population (OR 1.15; 95% CI 1.10–1.21), the subtropical group (OR 1.16; 95% CI 1.10–1.23), and the tropical group (OR 1.15; 95% CI 1.11–1.19) (Table 3).

### 3.3. Logistic Regression Model on the Association of the Cold Season with AMI Hospitalizations

The incidence of AMIs in the cold season was significantly and strongly associated with the cases aged ≥65 years relative to the cases aged <65 years in the total population (OR1.20; 95% CI 1.07 to 1.35) and the subtropical group (OR1.28; 95% CI 1.11 to 1.48). Similarly, the cold season incidence of AMI was significantly associated with those co-morbid for DM relative to the non-DM cases in the total population (OR 1.17; 95% CI 1.05 to 1.30) and in the tropical group (OR 1.25; 95% CI 1.04 to 1.51) (Table 4).

Further analysis of the ≥65 years aged group, within the cold season time period, showed a significant association with DM (a prevalence of 57.0%) relative to non-DM (a prevalence of 50.5%) in the tropical group (OR 1.32; 95% CI 1.04 to 1.67). No significant associations of cold season AMIs with potential confounders were found among the cases aged <65 years (Table 5).

Continued analysis within the DM group showed a significantly weak association with dyslipidemia during the cold season for AMI incidence (49.8%) relative to non-dyslipidemia (57.8%) in the tropical group (OR 0.74; 95% CI 0.55 to 0.99). Among the cases without DM, cold season AMI was significantly strong in association with dyslipidemia (58.3%) relative to non-dyslipidemia (49.8%) in the tropical group (OR 1.45; 95% CI 1.01 to 2.09) (Table 6).

## 4. Discussion

The main findings of this study suggest that the cold season incidence of AMI increased significantly compared to the hot season on the island of Taiwan, regardless of the subtropical and tropical regions. The incidence of AMI hospitalizations in the cold season was strongly associated with advanced age (≥65 years), especially in the subtropical region, and for cases with DM in the tropical region. Among the DM cases, a strong association was found within the non-dyslipidemia group, leading to a higher frequency of AMI during the cold season, but not for DM presenting with dyslipidemia. By contrast, among non-DM cases, the cold season AMI incidences were strongly associated with the presence of dyslipidemia relative to the cases without dyslipidemia.

The people in Taiwan exist as a homogeneous culture across the island and are included in the compulsory health insurance system of medical care in all regions of Taiwan. Taiwan is located in the South China sea and is considered an Asian population which shares the consistent staple of rice as the main component of the daily diet. Similarly, the gender ratios, average age of onset among men and women, prevalence of hypertension, DM, and dyslipidemia were consistent among the AMI cases in the subtropical and tropical regions. In a previous study from Taiwan, an absence of seasonal variation in the occurrences of AMI was claimed for a region lacking temperature extremes [10]. However, the present study found that seasonal variation in AMI hospitalizations is present, with a high rate during the cold season in Taiwan. Possible explanations for the discrepancy in the seasonal variation in AMI occurrences between these studies may be a result of different sample sizes and geographic scope. The current study investigated 7106 cases from across the island of Taiwan, while the Ku study relied on data from only two hospitals and 540 cases. In a recent study of Taiwan and Hong Kong, researchers identified an increase in the incidence of AMI as triggered below a temperature of 24 °C [11]. In Taiwan, the mean daily temperatures during the cold season descend down below 24 °C in November and ascend up to and over 24 °C by the end of April. Correspondingly, we found a greater incidence of AMI during the cold season as opposed to the hot season in both subtropical and tropical regions of Taiwan.

The effect of age and its association with season and AMI are stronger among older adults than young adults, and this finding is consistent in multiple studies [6,14,15]. However, some studies showed no differences [16]. In the study by Goggins et al. of Taiwan and Hong Kong, the older age groups showed a stronger association with the cold season and increased incidence of AMI compared to the younger age groups [11]. The present study is in agreement with these findings. The results of the advanced age group (≥65 years) and the increased incidence of AMI during the cold season showed a marked consistency in the subtropical region of Taiwan as well. The aging process is extremely complex and beyond the scope of this research, however, it is a well-established fact in the literature that decline in health is a result. Accordingly, the multiple functions of the skin, peripheral blood vessels, skeletal muscles, neural cells, and homeostatic mechanisms that maintain the body’s internal environment decrease in efficiency. The result of this inefficiency leads to the reduced capacity to adapt to changes in the external environment [17]. It is reasonable to suggest that the relatively high incidence of AMI among the ≥65 years cohort during the cold season is related to the decline in the body’s ability to adapt to temperature changes. This effect is more significant in the subtropical region of Taiwan, and is supported by the analysis.

DM is currently increasing in the global population and Taiwan is showing the same effect [18]. Extreme temperature has been shown to be a complicating factor in terms of hospital admissions and increased mortality in DM patients [19,20,21,22]. In temperate climate regions, previous studies showed no direct effect of DM and temperature related incidences of AMI [23,24]. However, in tropical climate regions, a study by Lam et al. of Hong Kong revealed that increased incidences of AMI during extreme temperatures were higher among DM patients compared to non-DM patients [11]. The temperature confounder showed an influence on the incidence of AMI, which is in agreement with the present study. More than half of AMI hospitalizations occurred during the cold season among both DM and non-DM patients, and the increased AMI hospitalizations during the cold season were strongly associated with DM comorbidity, reaching a significant level in the tropical region, particularly for the ≥65 years cohort. Furthermore, the discrepancy in the association of AMI hospitalizations during the cold season between the DM and non-DM groups was attributed to the difference between the DM group lacking dyslipidemia (57.8% in the tropical region versus 56.0% in the subtropical region) and the non-DM group lacking dyslipidemia (49.8% in the tropical region versus 51.7% in the subtropical region), rather than between DM patients with dyslipidemia (49.8% of occurrences in the tropical region versus 53.2% in the subtropical region) and non-DM patients with dyslipidemia (58.3% in the tropical region versus 53.8% in the subtropical region). The slight increase in the incidence of AMI among DM patients during the cold season and its association with the lack of, or presence of, dyslipidemia requires further investigation.

Research has shown strong links between DM, dyslipidemia, and increased rates of cardiovascular disease [25,26]. The interaction of the glucose and lipid metabolic dysfunction appears to exacerbate cardiovascular disease onset and incidence. The present study showed that DM patients without dyslipidemia and non-DM patients with dyslipidemia appeared to be more susceptible to AMI during the cold season, but DM patients with dyslipidemia and non-DM patients without dyslipidemia were equally susceptible, regardless of the season, and more markedly in the tropical region. On the basis of the above, it is reasonable to suggest that the degree of metabolic deterioration acts to modify the incidences of AMI between the cold and hot seasons. Metabolic function in non-DM patients without dyslipidemia showed no such susceptibility to an increased rate of AMI in association with temperature triggers, which suggests that the capacity to adapt was not impaired as it is in those afflicted. The findings by Lam et al. in a Hong Kong study showed hospitalization for AMI among the non-DM patients as increasing slightly as the ambient temperature dropped below 22 °C during the cold season, but no such association with temperature occurred during the hot season [12]. Similarly, the average daily temperature distribution showed fewer days with temperatures below 22 °C in the tropical region than in the subtropical region. Consequently, non-DM patients without dyslipidemia showed a tendency to be equally susceptible to AMI in both seasons, and more markedly in the tropical region. Systemic dysfunction due to metabolic deterioration in DM patients lacking dyslipidemia and non-DM patients with dyslipidemia did not appear to exert as negative an effect during the hot season, but was unable to counteract the stress of more extreme temperature drops (below 22–24 °C) during the cold season. However, systemic dysfunction due to DM patients who were co-morbid with dyslipidemia was too severe to counteract the stress of extreme temperatures during both the hot and cold seasons. As such, AMI hospitalizations among DM patients without dyslipidemia and non-DM patients with dyslipidemia were more frequent during the cold season than during the hot season, while DM patients with dyslipidemia were similar across both seasons. Of note in the findings by Lam et al. in their Hong Kong study, AMI hospitalization among the patients with DM increased markedly when temperatures dropped below 24 °C during the cold season and rose above 28.8 °C during the hot season [12]. The average daily temperature distribution showed more days with temperatures above 28.8 °C during the hot season in the tropical region than in the subtropical region. Consequently, the increase in AMI as a potential result of comorbid metabolic deterioration and dysfunction was more notable in the tropical region, in which the average mean daily temperature was only 1~2 °C higher than the subtropical region.

The treatment of AMI patients in Taiwan would follow the guidelines of the Taiwan Society of Cardiology, that were developed based on the guidelines of ACC/AHA for the management of patients with AMI. These guidelines emphasize the importance of primary prevention of acute coronary syndrome by decreasing long-term risk factors of coronary artery disease and avoiding potential short-term triggers. Risk awareness is a crucial aspect of preventative healthcare education. The identification of health specific triggers within a susceptible subset of the population is important in order to educate high-risk persons of their potential for negative effects. While the influence of weather and temperature remain beyond individual control, particularly increasing heatwaves following climate changes, it is possible to obtain predictive information with regards to temperature changes, allowing for changes in response, leading to reduced cardiovascular diseases [3]. According to potential strategies proposed by Tofler and Muller for acute risk prevention, reducing the absolute baseline risks caused by chronic health risk factors in susceptible persons needs to be the focus in order to both prepare and educate them with regard to the impact of specific triggers [1]. For instance, in ≥65 years persons at high risk for AMI and patients with either DM or dyslipidemia, it is suggested to engage in: (1) weight management, (2) cigarette smoking cessation, (3) moderate alcohol consumption, (4) increased physical activities, and (5) remain medication compliant on a consistent basis. Modification or avoidance of the triggers is another important strategy proposed by Tofler and Muller. People at higher risk for AMI should ensure adequate indoor temperatures and limit time outdoors during periods of extreme temperatures [1]. DM patients with dyslipidemia should be advised to take more precautionary measures to modify living environments against thermal stress, not only during periods of low temperature below 24 °C in the cold season but also high temperatures above 28.8 °C in the hot season.

This study has some strengths and limitations. A major strength of the current study is the use of the population-based database, the NHIRD, established by the National Health Insurance, which covers 99% of residents in Taiwan. Data of co-morbidities can be obtained from the longitudinal health insurance datasets of the NHIRD. The NHIRD can be used to analyze differences between seasons and climate zones of Taiwan. Limitations of the study include: the NHIRD is restricted to patients admitted to hospital and uses the principle diagnosis at discharge to identify AMI cases; diagnosis errors cannot be estimated; the NHIRD did not have information related to psychological or physical stress in response to changes in the cold and hot seasons; the restricted data source limited further investigations on the association between season-related risks for AMI and psychological or physical stress, seasonally; the island climate of Taiwan does not allow for the generalization of these results to other climatic regions.

## 5. Conclusions

The cold season shows an increased risk for AMI on the island of Taiwan. This is marked among the ≥65 years cohort in the subtropical region. However, the increased risks for AMI in the cold season are largely attributed to the contribution from DM patients lacking dyslipidemia and non-DM patients diagnosed with dyslipidemia, but not from the patients diagnosed with or without both DM and dyslipidemia in the tropical region. Age tends to influence season-related incidences of AMI in the relatively cool region, while comorbidity of metabolic dysfunction has an influence in the relatively warm region. As global warming is getting serious and metabolic disorders are increasing, further studies investigating the possible effects of climate change on season-related risks for AMI among patients with metabolic dysfunction are needed. The identification of temperature triggers for AMI in susceptible subsets of the population specifically may allow for effective preventative education measures to reduce the occurrences of AMI.

## Figures and Tables

**Figure 1 ijerph-16-02769-f001:**
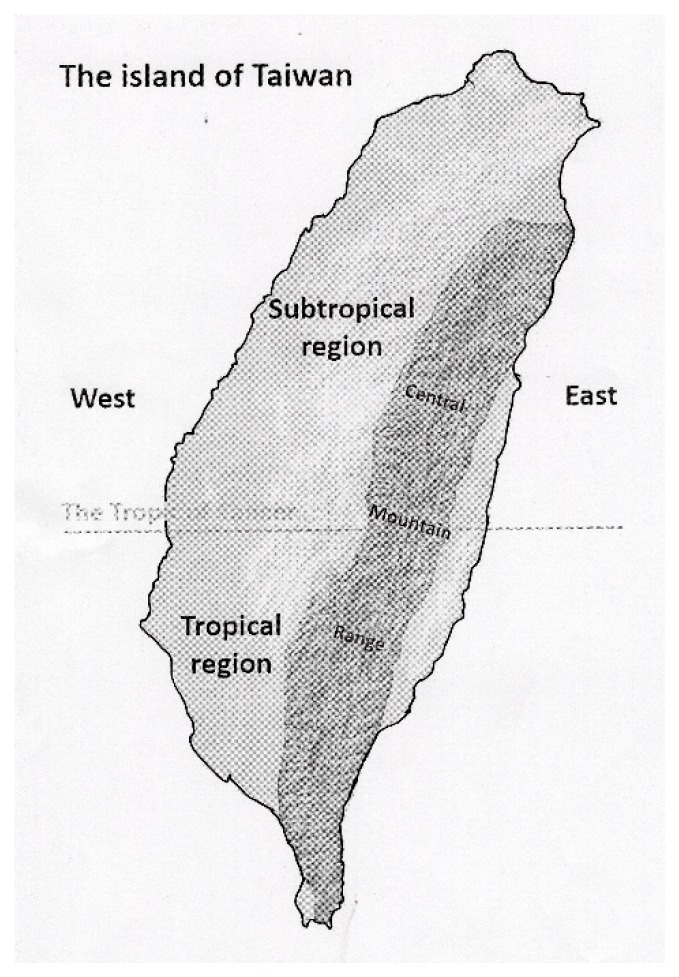
The subtropical and tropical regions of Taiwan. The main island of Taiwan is divided geographically into western and eastern areas by the central mountain range and climatically into tropical and subtropical regions by the Tropic of Cancer (23°26′12.6″ north of the Equator).

**Table 1 ijerph-16-02769-t001:** Summary of mean daily temperature from January 1997 to December 2011 in Taiwan.

	Cold Season			Hot Season		
	Nov	to	Apr	May	to	Oct
Subtropical region						
Monthly temperature (Mean ± SD) (°C)	21.9 ± 0.9	-	22.6 ± 1.2	25.7 ± 0.9	-	25.0 ± 0.9
Seasonal temperature (Mean ± SD) (°C)		19.2 ± 2.7			27.3 ± 1.8	
Tropical region						
Monthly temperature (Mean ± SD) (°C)	23.6 ± 1.1	-	25.2 ± 0.9	27.4 ± 0.6	-	26.5 ± 0.8
Seasonal temperature (Mean ± SD) (°C)		21.5 ± 2.7			28.0 ± 1.1	

SD: abbreviation of standard deviation.

**Table 2 ijerph-16-02769-t002:** Descriptive statistics of study population.

	Total				Subtropical Group				Tropical Group			
	*n* = 6897	%	Mean	±SD	*n* = 4604	%	Mean	±SD	*n* = 2293	%	Mean	±SD
Gender												
Male	4741	68.7			3152	68.5			1589	69.3		
Female	2156	31.3			1452	31.5			704	30.7		
Age							66.0	14.1			66.5	13.7
Male			63.9	13.9			63.6	13.9			64.4	13.8
Female			71.3	12.8			71.3	13.0			71.3	12.2
<65 years	3047	44.2			2065	44.9			982	42.8		
≥65 years	3850	55.8			2539	55.1			1311	57.2		
Comorbidity												
Hypertension	4286	62.1			2847	61.8			1439	62.8		
DM	2436	35.3			1580	34.3			856	37.3		
Dyslipidemia	1254	18.2			844	18.3			410	17.9		

SD: abbreviation of standard deviation; DM: abbreviation of Diabetes mellitus.

**Table 3 ijerph-16-02769-t003:** Comparison of acute myocardial infarctions (AMI) hospitalizations between the cold and hot seasons.

Cold–Hot Seasons	Cold Season						Hot Season						OR (95%CI)	*p*-Value
	Nov	Dec	Jan	Feb	Mar	Apr	May	Jun	Jul	Aug	Sep	Oct		
Total														
No. of AMI Hosp. each month	576	612	661	620	590	580	567	492	543	572	523	561		
No. of AMI Hosp./No. of Overall Hosp. (%) in Total	3639/242,791 (1.5%)						3258/249,572 (1.3%)						1.15 (1.10–1.21)	<0.001 ^★^
Subtropical group														
No. of AMI Hosp. each month	374	397	437	408	411	412	373	333	369	375	349	366		
No. of AMI Hosp./No. of Overall Hosp. (%) in Total	2439/171,304 (1.4%)						2165/176,298 (1.2%)						1.16 (1.10–1.23)	<0.001 ^★^
Tropical group														
No. of AMI Hosp. each month	202	215	224	212	179	168	194	159	174	197	174	195		
No. of AMI Hosp./No. of Overall Hosp. (%) in Total	1200/71,847(1.7%)						1093/73,274 (1.5%)						1.15 (1.11–1.19)	0.040 ^★^

No.: abbreviation of numbers; Hosp.: hospitalizations; ^★^ indicates *p* < 0.05.

**Table 4 ijerph-16-02769-t004:** Logistic regression model on the association of the cold season with AMI hospitalization.

	Total *n* = 6897			Subtropical Group *n* = 4604			Tropical Group *n* = 2293		
	Event n/N (%)	OR (95%CI)	*p*-Value	Event n/N (%)	OR (95%CI)	*p*-Value	Event n/N (%)	OR (95%CI)	*p*-Value
Age									
<65 years (ref.)	1539/3047 (50.5)	1		1037/2065 (50.2)	1		502/982 (51.1)	1	
≥65 years	2100/3850 (54.5)	1.20 (1.07–1.35)	0.002 ^★^	1402/2539 (55.2)	1.28 (1.11–1.48)	0.001 ^★^	698/1311 (53.2)	1.06 (0.87–1.30)	0.165
Gender									
Male (ref.)	2499/4741 (52.7)	1		1659/3152 (52.6)	1		840/1589 (52.9)	1	
Female	1140/2156 (52.9)	1.01 (0.84–1.20)	0.951	780/1452 (53.7)	1.10 (0.88–1.37)	0.399	360/704 (51.1)	0.80 (0.58–1.10)	0.165
Age by Gender		0.93 (0.75–1.16)	0.527		0.84 (0.64–1.10)	0.202		1.17 (0.90–1.72)	0.427
Diabetes mellitus									
No	2297/4461 (51.5)	1		1570/3024 (51.9)	1		727/1437 (50.6)	1	
Yes	1342/2436 (55.1)	1.17 (1.05–1.30)	0.005 ^★^	869/1580 (55.0)	1.12 (0.98–1.28)	0.111	473/856 (55.3)	1.25 (1.04–1.51)	0.018 ^★^
Hypertension									
No	1358/2611 (52.0)	1		913/1757 (52.0)	1		445/854 (52.1)	1	
Yes	2281/4286 (53.2)	0.95 (0.85–1.06)	0.329	1526/2847 (53.6)	0.97 (0.85–1.11)	0.674	755/1439 (52.5)	0.94 (0.78–1.14)	0.522
Dyslipidemia									
No	2972/5643 (52.7)	1		1988/3760 (52.9)	1		984/1883 (52.3)	1	
Yes	667/1254 (53.2)	0.97 (0.85–1.10)	0.584	451/844 (53.4)	0.98 (0.83–1.15)	0.784	216/410 (52.7)	0.97 (0.77–1.21)	0.764

ref.: abbreviation of reference; ^★^ indicates *p* < 0.05.

**Table 5 ijerph-16-02769-t005:** The association of the cold season with AMI hospitalization among the ≥65 years group and the <65 years group.

**≥65 Years Group**	**Subtropical Group *n* = 2539**			**Tropical Group *n* = 1311**		
**N = 3850**	**Event n/N (%)**	**OR (95%CI)**	***p*-Value**	**Event n/N (%)**	**OR (95%CI)**	***p*-Value**
Gender						
Male (ref.)	828/1480 (55.9)	1		430/802 (53.6)	1	
Female	574/1059 (54.2)	0.93 (0.79–1.09)	0.346	268/509 (52.7)	0.92 (0.73–1.15)	0.453
Diabetes mellitus						
No	518/1495 (54.5)	1		382/757 (50.5)	1	
Yes	587/1044 (56.2)	1.10 (0.93–1.31)	0.257	316/554 (57.0)	1.32 (1.04–1.67)	0.022 ^★^
Hypertension						
No	338/610 (55.4)	1		166/324 (51.2)	1	
Yes	1064/1929 (55.2)	0.98 (0.81–1.19)	0.859	532/987 (53.9)	1.04 (0.80–1.35)	0.781
Dyslipidemia						
No	1118/2017 (55.4)	1		566/1068 (53.0)	1	
Yes	284/522 (54.4)	0.94 (0.77–1.15)	0.556	132/243 (54.3)	0.96 (0.72–1.29)	0.799
**<65 Years Group**	**Subtropical Group *n* = 2065**			**Tropical Group *n* = 982**		
**N = 3047**	**Event n/N (%)**	**OR (95%CI)**	***p*-Value**	**Event n/N (%)**	**OR (95%CI)**	***p*-Value**
Gender						
Male (ref.)	831/1672 (49.7)	1		410/787 (52.1)	1	
Female	206/393 (52.4)	1.10 (0.88–1.37)	0.421	92/195 (47.2)	0.82(0.60–1.14)	0.236
Diabetes mellitus						
No	755/1529 (49.4)	1		345/680 (50.7)	1	
Yes	282/536 (52.6)	1.14 (0.91–1.42)	0.260	157/302 (52.0)	1.16(0.85–1.56)	0.352
Hypertension						
No	575/1147 (50.1)	1		279/530 (52.6)	1	
Yes	462/918 (50.3)	0.95 (0.79–1.15)	0.619	223/452 (49.3)	0.86(0.65–1.13)	0.274
Dyslipidemia						
No	870/1743 (49.9)	1		418/815 (51.3)	1	
Yes	167/322 (51.9)	1.04 (0.80–1.35)	0.767	84/167 (50.3)	0.98(0.69–1.41)	0.928

ref.: abbreviation of reference; ^★^ indicates *p* < 0.05.

**Table 6 ijerph-16-02769-t006:** The association of the cold season with AMI hospitalization among the DM group and the non-DM group.

**DM Group**	**Subtropical Group *n* = 1580**			**Tropical Group *n* = 856**		
***n* = 2436**	**Event n/N (%)**	**OR (95%CI)**	***p*-Value**	**Event n/N (%)**	**OR (95%CI)**	***p*-Value**
Age						
<65 years (ref.)	282/536 (52.6)	1		157/302 (52.0)	1	
≥65 years	587/1044 (56.2)	1.20 (0.92–1.57)	0.177	316/554 (57.0)	1.10 (0.77–1.58)	0.605
Gender						
Male (ref.)	497/907 (54.8)	1		270/501 (53.9)	1	
Female	372/673 (55.3)	1.06 (0.73–1.55)	0.758	203/355 (57.2)	0.96 (0.58–1.59)	0.878
Age by Gender		0.92 (0.59–1.45)	0.733		1.26 (0.69–2.30)	0.455
Hypertension						
No	132/231 (57.1)	1		78/139 (56.1)	1	
Yes	737/1349 (54.6)	0.88 (0.66–1.18)	0.402	395/717 (55.1)	0.94 (0.64–1.37)	0.749
Dyslipidemia						
No	572/1022 (56.0)	1		338/585 (57.8)	1	
Yes	297/558 (53.2)	0.91 (0.74–1.12)	0.386	135/271 (49.8)	0.74 (0.55–0.99)	0.040 ^★^
**Non-DM Group**	**Subtropical Group *n* = 3024**			**Tropical Group *n* = 1437**		
***n* = 4461**	**Event n/N (%)**	**OR (95%CI)**	***p*-Value**	**Event n/N (%)**	**OR (95%CI)**	***p*-Value**
Age						
<65 years (ref.)	755/1529 (49.4)	1		345/680 (50.7)	1	
≥65 years	815/1495 (54.5)	1.31 (1.10–1.56)	0.002^★^	382/757 (50.5)	1.07 (0.83–1.37)	0.610
Gender						
Male (ref.)	1162/2245 (51.8)	1		570/1088 (52.4)	1	
Female	408/779 (52.4)	1.13 (0.86–1.48)	0.394	157/349 (45.0)	0.75 (0.49–1.14)	0.174
Age by Gender		0.79 (0.56–1.11)	0.175		0.99 (0.59–1.67)	0.975
Hypertension						
No	781/1526 (51.2)	1		349/715 (51.3)	1	
Yes	789/1498 (52.7)	0.99 (0.85–1.15)	0.875	360/722 (49.9)	0.91 (0.73–1.14)	0.409
Dyslipidemia						
No	1416/2738 (51.7)	1		646/1298 (49.8)	1	
Yes	154/286 (53.8)	1.08 (0.84–1.39)	0.535	81/139 (58.3)	1.45 (1.01–2.09)	0.044 ^★^

ref.: abbreviation of reference; ^★^ indicates *p* < 0.05.

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
