# Peer review of "Acute Myocardial Infarction Hospitalizations between Cold and Hot Seasons in an Island across Tropical and Subtropical Climate Zones—A Population-Based Study"

_ijerph, 2019, doi:10.3390/ijerph16152769_

Round 1

Reviewer 1 Report

In this paper the authors show that for inhabitants of Taiwan, there is a higher risk for hospitalization due to AMI during the cold half year than in the hot half year. This  risk is higher for older than younger people, and for DM2 patients than nonDM2 patients. Unfortunately, data on monthly AMI hospitalizations are not expressed relative to overall hospitalizations or overall mortality.

Major.

From ref #2 it follows that there is a much bigger difference in overall mortality between the cold and hot season in Taiwan than found in the current paper. If this also holds for hospitalizations, the conclusion may well be that people in Taiwan are actually protected against AMI hospitalizations during the cold season. Data on AMI hospitalizations should therefore be expressed relative to overall hospitalizations. 

According to ref #4, the frequency of AMI hospitalizations is highly related to seasonal changes or variability in temperature. Yet, temperature data are given as average daily or monthly temperatures rather than temperature variiabities.. 

Minor: 

In the abstract the time period of data collection is missing. Further, the OR (95%CI) for AMI hospitalization in cold vs hot seacon should be included in the abstract. 

Results: lines 115-117 ?

Table 2: the columns are noet aligned properly

Author Response

Response to Reviewer 1 Comments

Major

Point 1: From ref #2 it follows that there is a much bigger difference in overall mortality between the cold and hot season in Taiwan than found in the current paper. If this also holds for hospitalizations, the conclusion may well be that people in Taiwan are actually protected against AMI hospitalizations during the cold season. Data on AMI hospitalizations should therefore be expressed relative to overall hospitalizations. 

Response 1: Thank you for your comments and suggestions. As suggested, the incidence of AMI hospitalizations was redefined as number of AMI hospitalizations divided by number of overall hospitalizations. The incidence (1.5% vs 1.3%) and odds ratio (OR 1.15; 95% CI 1.10- 1.21) showed a significantly higher risk of AMI hospitalizations during the cold season compared to the hot season. The results were updated in Table 3 and Line 142-145.

Point 2: According to ref #4, the frequency of AMI hospitalizations is highly related to seasonal changes or variability in temperature. Yet, temperature data are given as average daily or monthly temperatures rather than temperature variiabities.

Response 2: Point taken. For showing variability in temperature, the data of seasonal temperature were express as mean ± standard deviation (SD). The results were updated in Table 1 and Line 116-118.

Minor

Point 1: In the abstract the time period of data collection is missing. Further, the OR (95%CI) for AMI hospitalization in cold vs hot seacon should be included in the abstract.

Response 1: Thank you for your comments. The time period of data collection was added (Line 17) and the odds ratio (OR 1.15; 95% CI 1.10- 1.21) was provided (Line 22) in the abstract accordingly.

Point 2: Results: lines 115-117 ?

Response 2: Point taken and explained. The sentences of lines 115-117 were originally in the Microsoft Word template provided by this journal and supposed to be deleted. The deletion of these sentences were done.

Point 3: Table 2: the columns are noet aligned properly

Response 3: Thanks for your reminding. The realignment was done in Table 2.

Reviewer 2 Report

It would be good if the authors would let the location of Taiwan in the methods section.

Could the authors make any comment regarding the treatment that their patients received?

Author Response

Response to Reviewer 2 Comments

Point 1: It would be good if the authors would let the location of Taiwan in the methods section.

Response 1: Thank you for your suggestions. The sentence describing the location of Taiwan in the introduction section (Line 61) was relocated to the method section (Line 82).

Point 2: Could the authors make any comment regarding the treatment that their patients received?

Response 2: Thank you for your suggestions. The treatment of AMI patients in Taiwan would follow the guidelines of the Taiwan Society of Cardiology that was developed based on the guidelines of ACC/AHA for the management of patient with AMI. These guidelines emphasize the importance of primary prevention of acute coronary syndrome by decreasing long-term risk factors of coronary artery disease and avoiding potential short-term triggers. The sentences above were added in the discussion section (Line 278-282).

Reviewer 3 Report

In this manuscript titled, "Acute Myocardial Infarction Hospitalizations between Cold and Hot Seasons in an Island across Tropical and Subtropical Climate Zones – A population-based study", Chu-Min Liang et al., authors investigated the effects of cold and hot seasons on hospital admissions of acute myocardial infarction (AMI) at the junction of tropical and subtropical climate zones on the island of Taiwan. Authors revealed that the cold season incidence of AMI increased significantly compared to the hot-season, regardless of subtropical and tropical regions of Taiwan. This study may allow for effective preventive education measures to decrease the occurrences of AMI at regions of Taiwan.  Overall, the manuscript is written clearly.  However, the manuscript appears preliminary.

1.     p-value: p should be upper case and italic ‘  P ’, in this manuscript authors use both P and p.

2.     p<.05  under table 3-6 should be P<0.05

3.     In the discussion, line 194 and 203 authors should use ‘the island of Taiwan or regions of Taiwan’ to replace ‘nationwide and nation’.The island climate of Taiwan does not allow for generalization of these results to other climatic regions of the whole country of China.

Author Response

Response to Reviewer 3 Comments

Point 1: p-value: p should be upper case and italic ‘  ’, in this manuscript authors use both P and p.

Response 1: Thank you for your suggestions. The modification of P instead of P and p was done throughout this manuscript.

Point 2: p<.05  under table 3-6 should be P<0.05

Response 2: Thank you for your suggestions. The modification of P<0.05 instead of p<.05 was done in tables 3-6.

Point 3:  In the discussion, line 194 and 203 authors should use ‘the island of Taiwan or regions of Taiwan’ to replace ‘nationwide and nation’.The island climate of Taiwan does not allow for generalization of these results to other climatic regions of the whole country of China.

Response 3: Thank you for your comments and suggestions. ‘in all regions of Taiwan’ was used to replace ‘nationwide’ in line 197 and ‘the island of Taiwan’ was used to replace ‘nation’ in line 206.

Reviewer 4 Report

The study was conducted well enough. However, a small specter of confuses a little. This topic needs further investigations to define the factors, affecting the incidence of MI. For example, the exact temperature level itself may not be the driving factor for MI, but MI may be caused by the psychological or physical stress, due to the perception of this temperature levels as uncomfortable. In this case, the additional test of Anxiety levels may reveal some interesting interconnections.

In addition, I would suggest thinking about the influence of holiday-labor periods/cycle, which in turn may affect the incidence of MI. But this points are only suggestions for the future studies.

With that, it was a good work done. Thank you.

Author Response

Response to Reviewer 4 Comments

Point 1: The study was conducted well enough. However, a small specter of confuses a little. This topic needs further investigations to define the factors, affecting the incidence of MI. For example, the exact temperature level itself may not be the driving factor for MI, but MI may be caused by the psychological or physical stress, due to the perception of this temperature levels as uncomfortable. In this case, the additional test of Anxiety levels may reveal some interesting interconnections.

Response 1: Point taken. As mentioned in the introduction section (line 35-36), coronary artery occlusion may be influenced not only by long-term chronic physiological risk factors but also by short-term exposure to physical, psychosocial, and environmental triggers. However, the data source of the studied cases, NHIRD, did not have information related to psychological or physical stress in response to changes in cold-hot seasons. The restricted data source limited further investigations on the association between season-related risks for AMI and psychological or physical stress seasonally. This limitation of the study was added in line 306-308.

Point 2: In addition, I would suggest thinking about the influence of holiday-labor periods/cycle, which in turn may affect the incidence of MI. But this points are only suggestions for the future studies.

Response 2: Thank you for your suggestions. Many studies have provided evidences for the association between the incidence of MI and holiday-labor periods/circadian cycle. We would take this issues into consideration for the future studies.

Round 2

Reviewer 1 Report

The authors have responded adequately to my comments and suggestions. There are a few typo's that need to be corrected. 

Line 24: cold-season-AMIs; dysrlipidemia

Table 1: column header: toSep

Table 3: line with No of AMI/hospitalizations: 3,639/224,2791